# Quantum-dot and organic hybrid tandem light-emitting diodes with multi-functionality of full-color-tunability and white-light-emission

Heng Zhang[1], Qiang Su[1] & Shuming Chen [1✉]

Realizing of full-color quantum-dot LED display remains a challenge because of the poor stability of the blue quantum-dot and the immature inkjet-printing color patterning technology. Here, we develop a multifunctional tandem LED by stacking a yellow quantum-dot LED with a blue organic LED using an indium–zinc oxide intermediate connecting electrode. Under parallel connection and alternate-current driving, the tandem LED is full-color-tunable, which can emit red, green and blue primary colors as well as arbitrary colors that cover a 63% National Television System Committee color triangle. Under series connection and direct current driving, the tandem LED can emit efficient white light with a high brightness of 107000 cd m$^{-2}$ and a maximum external quantum efficiency up to 26.02%. The demonstrated hybrid tandem LED, with multi-functionality of full-color-tunability and white light-emission, could find potential applications in both full-color-display and solid-state-lighting.

[1] Department of Electrical and Electronic Engineering, Southern University of Science and Technology, 518055 Shenzhen, P. R. China. ✉email: chen.sm@sustech.edu.cn

CdSe-based quantum-dot light-emitting diodes (QLEDs) have been extensively explored for applications in display and lighting due to their unique merits of high color saturation, tunable emission color, high brightness, and simple solution processability[1–12]. Recent advances have enabled QLEDs to exhibit high external quantum efficiency (EQE) and long operational lifetime. For instance, the EQEs of the state-of-the-art red- (R-), green- (G-), and blue- (B-) QLEDs are higher than 20.5%[1,3,13,14], 22%[3,15–18], and 18%[14,19], respectively, while the $T_{50}$ lifetime of R- and G-QLEDs are longer than 26,500 (refs. [3,20]) and 25000 h (refs. [3,18]) at an initial brightness of 1000 cd m$^{-2}$. Although the efficiency of QLEDs could meet the requirements of display applications, the realization of QLED full-color display remains challenging.

One reason is that the B-QLEDs are unstable, with a short $T_{50}$ lifetime of ∼200 h at an initial brightness of 1000 cd m$^{-2}$ (refs. [3,21]), which lags far behind those of R- and G-QLEDs. While the stability of B-QLEDs is being questioned, its organic cousins—the fluorescent organic light-emitting diodes (FL-OLEDs), are relatively stable and have been applied in displays for years. By substituting the B-QLEDs with B-OLEDs, a hybrid display could be realized, which can enjoy both the high saturation of QLEDs as well as the high stability of OLEDs. However, promoting the marriage between QLEDs and OLEDs could be difficult, since they come from different families.

Another reason is that the inkjet printing, which is used to deposit and pattern the light-emitting layers (EMLs), has many shortcomings such as low resolution, poor uniformity, and is far from mature for mass-producing QLED displays[22–25]. Instead of patterning the EMLs directly, the R/G/B side-by-side color pixels can also be realized by combining the white devices with the patterned color filters (CF)[26,27]. The "white+CF" scheme has been proven to be an effective strategy for manufacturing large-area OLED televisions and high-resolution OLED microdisplays[26–29]. However, the introduction of CFs significantly reduces the brightness for more than 2/3 of the white light is absorbed by the CFs. To eliminate the absorptive CFs, it is desirable to develop a color-tunable device that can emit R, G, and B primary colors through the variation of driving conditions. By emitting the R, G, and B colors time-sequentially, a full-color image can be displayed. In this case, neither CFs nor EMLs patterning are needed. Moreover, the pixel density and fill factor of the display can be increased by threefold, since a single color-tunable pixel can take the job of three side-by-side R/G/B pixels, as shown in Supplementary Fig. 1. Driven by these benefits, color-tunable devices have long been pursued. In 1990s, Forrest and co-authors first obtained full-color-tunable OLEDs by vertically stacking R-, G-, and B-OLEDs[30,31], but it requires four independently addressable electrodes to drive these devices, which complicates the fabrication and the drive circuits. To make it practical for display application, one has to invent a new architecture with only two electrodes. Unfortunately, all reported two-terminal devices can only emit two primary colors at most[32–36], making them useless for full-color displays.

To address aforementioned challenges, we hereby develop a two-terminal and full-color-tunable device, which is realized by stacking a yellow (Y)-QLED with a B-OLED using an indium–zinc oxide (IZO) intermediate connecting electrode (ICE). The fluorescent organic molecules are adopted as the B-emitters to substitute the unstable B-QDs. With this novel tandem structure, QLED and OLED can be well integrated, either in parallel or in series, in a single two-terminal architecture. By varying the driving alternate-current (AC) signals, the device can emit R, G, and B primary colors as well as arbitrary colors, making it an ideal candidate for CF-free and EML-patterning-free full-color displays. Besides full-color-tunability, the device can also

emit efficient white light with a high brightness of 107000 cd m$^{-2}$ and an EQE up to 26.02% when under direct-current (DC) driving. With a novel AC driving, the white LED can emit stable colors with 1931 Commission International de l'Eclairage (CIE) coordinates fixed at (0.34, 0.36) over a wide range of brightness (1000–50,000 cd m$^{-2}$). Also, the color coordinates can be tuned to trace the blackbody locus over a wide range of correlated color temperature (1500–10,000 K). We believe that our demonstrated hybrid tandem LED, with multi-functionality of full-color tunability and white light emission, could find potential applications in both full-color display and solid-state lighting.

## Results

**Device structure and functionalities.** As shown in Fig. 1, the multifunctional tandem LED consists of a bottom CdSe-based Y-QLED and a top MADN:DSA-Ph-based FL B-OLED, which are vertically stacked and connected through an ICE. To obtain higher efficiency, the FL B-OLED can be replaced by a mCPPO1:FIrPic-based phosphorescent (Ph) B-OLED. Compared with conventional tandem devices that usually employ p/n hetero-layers as the connection media[14,17,37,38], our device adopts a conductive and transparent IZO as the ICE, which can enable an effective electrical connection as well as an efficient optical coupling. Also, the IZO could promote the hybrid marriage between QLED and OLED. As shown in Fig. 1b, with IZO as the ICE, charge carriers can be efficiently injected into both Y-QLED and B-OLED, leading to the efficient light emission. Moreover, the IZO can be extracted as an independent electrode, thereby allowing us to control the devices in different ways, as will be discussed later.

The IZO was deposited by sputtering. To protect the quantum-dot EML from ion bombardment damage during sputtering process, an ultra-thin (2 nm) Al was introduced, which is previously demonstrated as an effective buffer layer for transparent QLED[39]. Coupled with the protection of ZnMgO electron transport layer, the bombardment damage could be minimized[40]. Indeed, as shown in Supplementary Fig. 2, with the sputtered 80-nm IZO as the top electrode, the transparent R-QLED exhibits an EQE of 11.4%, which is slightly smaller than 12.5% of the conventional device, implying the good feasibility of using IZO as the top electrode for damage-free QLED.

The adoption of IZO as intermediate electrode, which enables the devices to be connected either in series or in parallel, is a key strategy for the realization of multifunctional tandem LED. Conventionally, with bottom ITO as an anode and top Al as a cathode, the B-OLED and Y-QLED are connected in series, and under the driving of a DC source, efficient white light emission can be obtained. Also, the devices can be connected in parallel by combing the bottom ITO and the top Al as one electrode, and extracting the IZO as the counter electrode, as shown in Fig. 1a. In this way, the Y-QLED is reversely biased while the B-OLED is forwardly driven, and thus, both devices can be independently addressed by the polarity of the driving signals, consequently enabling a color-tunable device, as will be discussed in the next section.

**Full-color-tunable LED.** Under parallel connection and AC driving, the Y-QLED is turned-on by the negative pulses while the B-OLED is lit-up by the positive ones, and thus the tandem LED alternately emits the Y and B colors. To display the full-color, a third color should be introduced. To this end, the Y-QLED is specially configured, with its EML consisting of mixing R- and G-QDs. As shown in Fig. 2a, at a small driving voltage, most charge carriers tend to inject and recombine on the R-QDs due to their small injection barrier. The exciton recombination zone could gradually migrate from the R-QDs to the G-QDs when the driving voltage is increased. Therefore, the emitting color of the Y-QLED

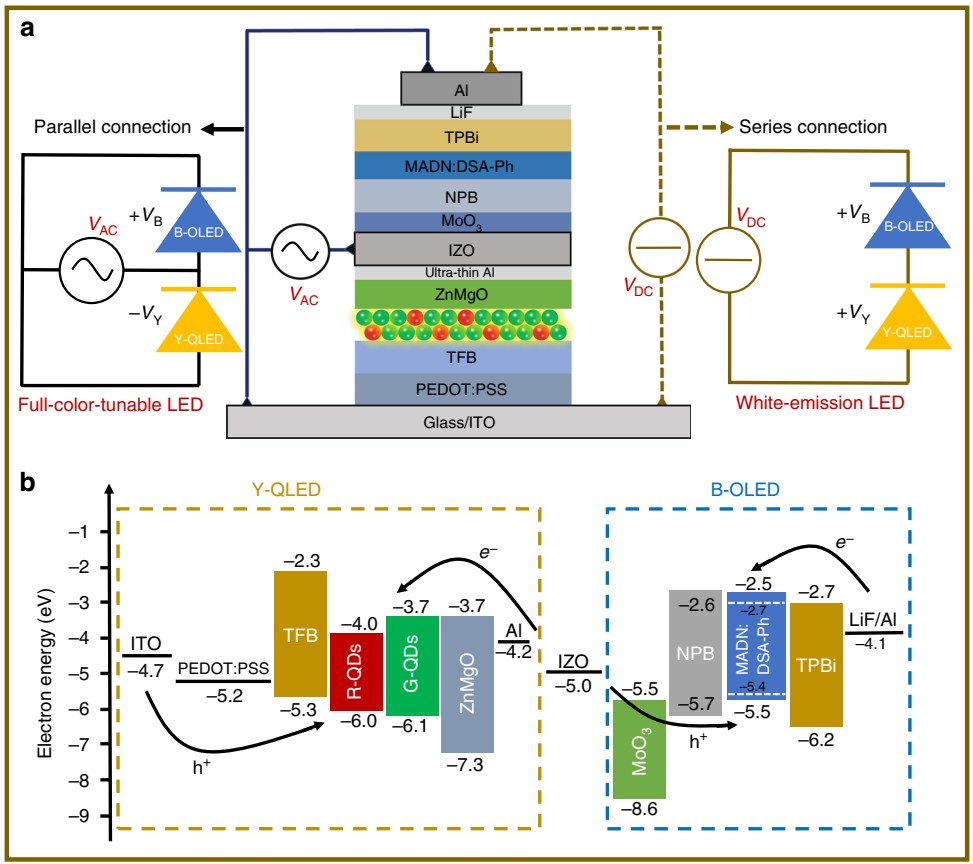

**Fig. 1 Device structure and driving methods of the multifunctional LED. a** Schematic device structure of the multifunctional tandem LED with an extractable IZO intermediate electrode. The Y-QLED and B-OLED can be connected either in parallel or in series. Under parallel connection and AC driving, a color-tunable device is achieved. Under series connection and DC driving, a white-emitting device is obtained. **b** Energy level alignment of the multifunctional tandem LED.

is controlled by the voltage, which can be tuned from pure R to pure G by increasing the voltage. With such configuration, the emitting color of the tandem LED can be controlled by both polarity and amplitude of the driving signals. Therefore, by simply varying the driving AC signals, the tandem LED can alternately emit three primary colors, with the B color controlled by the positive pulses, the R and the G colors activated by the amplitude of the negative pulses. Also, arbitrary colors, located inside a color triangle defined by the primary colors, can be obtained by integrating the emission of both Y-QLED and B-OLED.

Figure 2b–d shows the driving signals, the CIE chart, the photos, and the emission spectra of a color-tunable tandem LED that emits different colors. To drive the device, an AC source with positive pulses $V_B$ and negative pulses $V_Y$, which control the luminance of B-OLED and Y-QLED, respectively, was applied. The frequency of the AC source was set at 100 Hz so that human eyes perceive a combined emission from both B-OLED and Y-QLED. As shown in Fig. 2b, d, the color can be continuously tuned from:

(I)   R to G. In this case, only the Y-QLED is activated while the B-OLED is switched off by setting $V_B = 0$. By gradually increasing the $V_Y$ from −2 to −8 V, the emission color is tuned from R, Y to G. The photos and the emission spectra of the tandem LED with different colors are displayed in Fig. 2b and d, respectively. Pure R emission with a color coordinate of (0.67, 0.32) is obtained at 2 V, while the G emission is achieved at 8 V and its purity is affected by the blending ratio of R- and G-QDs. As shown in Supplementary Fig. 3, the purity of G is degraded by the R emission if

excess R-QDs are incorporated. The purity of G is also affected by the thickness of IZO due to the microcavity effect, as shown in Supplementary Fig. 4. At an optimal R-QDs:G-QDs mixing ratio of 1:20 and an IZO thickness of 60 nm, the purest G emission with a color coordinate of (0.26, 0.29) is obtained.

(II)   B to R. The B emission is obtained by activating the B-OLED only. When both B-OLED and Y-QLED are alternately turned-on, a new color resulting from the mixing of the emission of both devices is generated. To ensure a pure R emission, the $V_Y$ is set at −2 V. By decreasing the $V_B$ from 4 to 0 V, the blue intensity is gradually reduced and thus the color is tuned from B (0.16, 0.31), purple (0.27, 0.32) to R (0.67, 0.32).

(III)   B to orange (O). The driving method is similar to that of (II). Except that the $V_Y$ is set at −2.8 V to produce an O emission. Also, to balance the light intensities of both devices, the maximum $V_B$ is raised to 5 V and the duty ratio (DR) is increased to 80%. The color is changed from B (0.16, 0.31), warm white (0.37, 0.40) to O (0.52, 0.46) when the $V_B$ is decreased from 5.5 to 0 V.

(IV)   B to G. The driving method is similar to that of (III). Except that the $V_Y$ is set at −8 V to ensure a pure G emission, and the DR is further increased to 95%. When the $V_B$ is decreased from 10 to 0 V, the colors are shifted from B (0.16, 0.31), cyan (0.18, 0.40) to G (0.26, 0.29).

As shown in Fig. 2b, the colors that are achievable can cover a color triangle with a 63% NTSC (National Television System

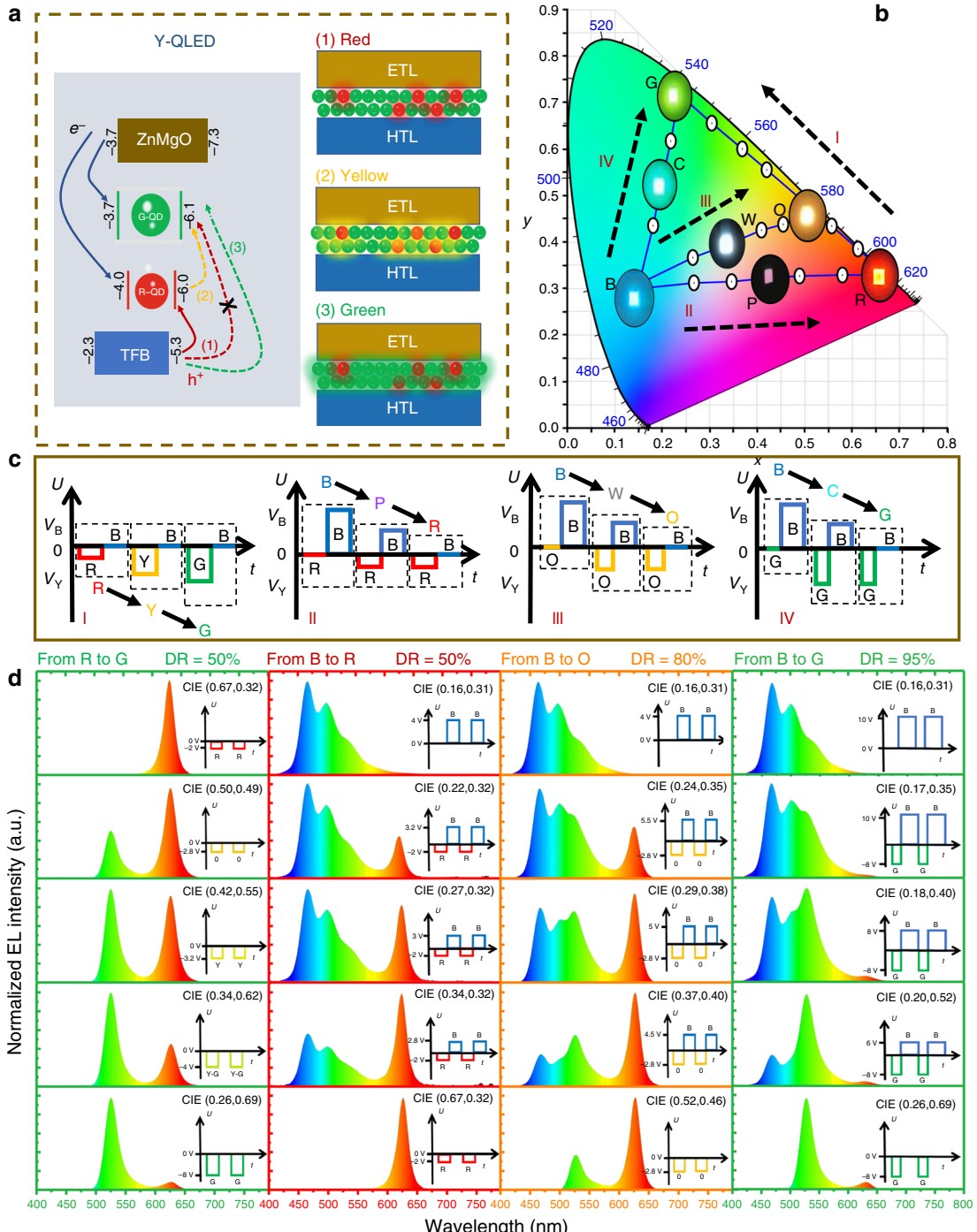

**Fig. 2 Mechanism and driving methods of full-color-tunable LED. a** Schematic diagram of the charge injection and recombination trajectory of the Y-QLED. The recombination zone is gradually migrated from R-QDs to G-QDs when the driving voltage is increased, and thus the emission color of Y-QLED can be controlled by the voltage. **b** The CIE chart, CIE coordinates, and photos; **c** the AC driving signals; and **d** the emission spectra of a color-tunable device that emits different colors. The color can be continuously tuned from (I) R to G, (II) B to R, (III) B to O, and (IV) B to G.

Committee) color gamut. To the best of our knowledge, this is the first demonstration of a full-color-tunable device that is simply driven by a single two-terminal source (also demonstrated in Supplementary Movie). The full-color-tunable device, which can take the job of three conventional R, G, and B sub-pixels, can enable a CF-free display with higher pixel density. To display a colorful picture, each frame can be divided into three sub-frames, corresponding to the R, G, and B colors of a picture. By fast and time-sequentially displaying the R, G, and B colors, a full-color picture can be synthesized. The gray level is also adjustable, since

the B intensity can be independently controlled by the $V_B$, while the R and G intensity can be altered by tuning the driving time.

The performance of B-OLED and Y-QLED can be separately evaluated by extracting the IZO electrode. As shown in Fig. 3a, a high EQE of 5.6% and 11.2% are achieved by B-OLED and Y-QLED, respectively, indicating that the introduction of IZO does not affect the performance adversely. The current density–voltage–luminance ($J$–$V$–$L$) characteristics of the devices are shown in Fig. 3b. The B-OLED and Y-QLED exhibit a low turn-on voltage of 2.8 and 2.1 V and a maximum brightness of

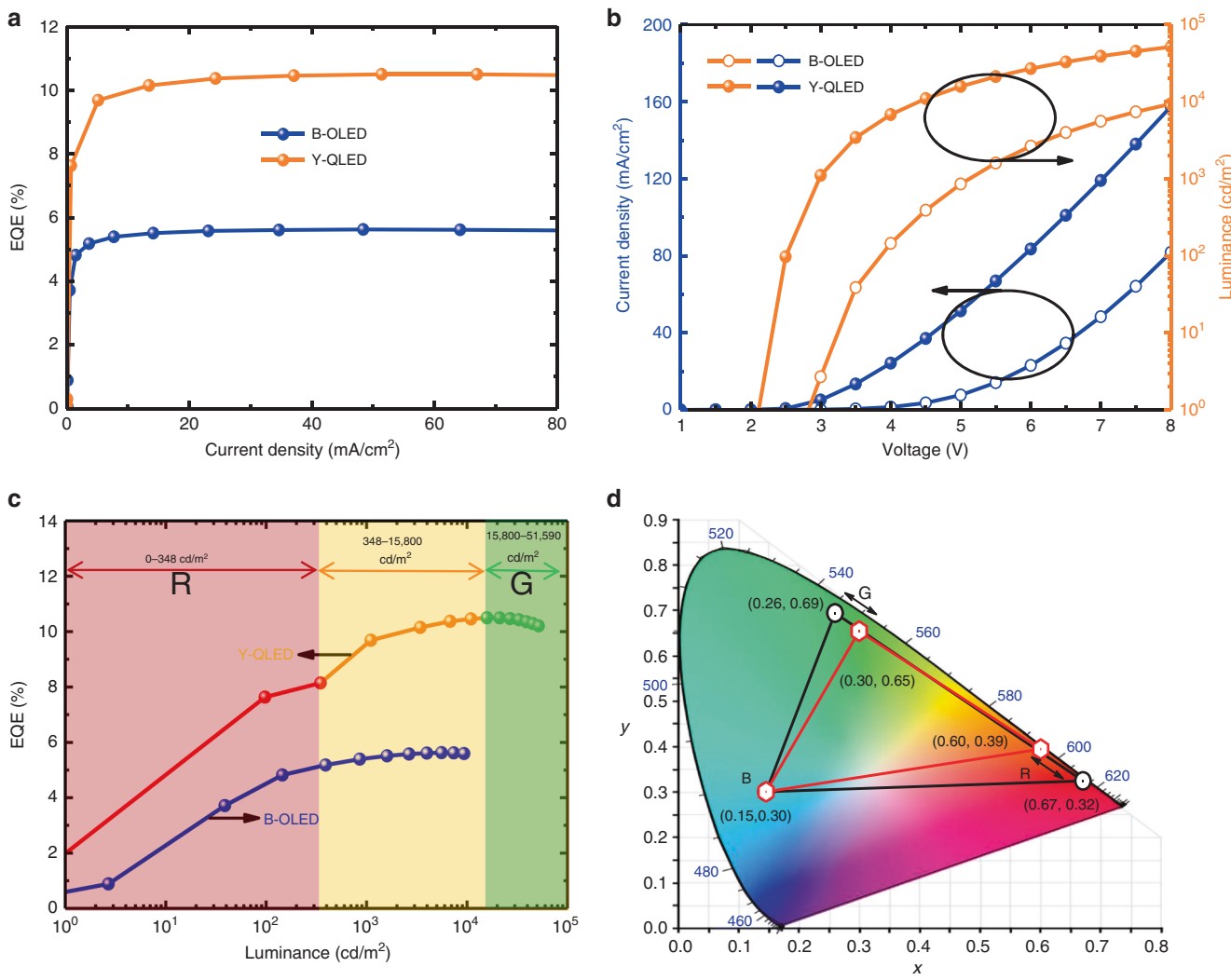

**Fig. 3 Performance of the full-color-tunable LED. a** The *EQE–J*, **b** *J–V*–L, and **c** *EQE–L* characteristics of B-OLED and Y-QLED, measured by extracting the IZO as a common electrode. **d** CIE chart and color coordinates of the R, G, and B primary colors. By degrading the color gamut from 63% to 45% NTSC, a wide range of brightness for the R, G, and B emission can be obtained.

9359 and 51590 cd m$^{-2}$, respectively. For the Y-QLED, the emission color is controlled by the driving voltage, and thus by correlating the emission color with the voltage, we can probe the brightness/efficiency of the R and G emission, as shown in Fig. 3c. The purest R emission with a CIE coordinate of (0.67, 0.32) is achieved at a low brightness of 0.35 cd m$^{-2}$, which is too low for practical application. By losing the color saturation and moving the coordinate to (0.60, 0.39), the brightness and EQE can be enhanced to 348 cd m$^{-2}$ and 8.2%, respectively, which could be sufficient for display application. Similarly, for the G emission, if we allow the color saturation degrading from (0.26, 0.69) to (0.30, 0.65), the brightness can be decreased from 51,590 to 15,800 cd m$^{-2}$, which could be further reduced to an application level by shorting the driving time in a frame period. In this case, the color gamut is decreased from 63% to 45% NTSC so that a wide range of brightness for the R, G, and B emission can be obtained, as shown in Fig. 3d. The color gamut can further be improved by enhancing the color saturation of the B-OLED. The detailed performances of the device are summarized in Table 1.

**Efficient and color stable white LED**. Under series connection and DC driving, both B-OLED and Y-QLED are turned-on

simultaneously. By integrating the emission from both devices, an efficient white emission is achieved. As shown in Fig. 4a, b, the tandem LED exhibits a turn-on voltage of 4.16 V, a brightness of 32575 cd m$^{-2}$ at 100 mA cm$^{-2}$, and a peak EQE of 16.91%, which are nearly equal to the sum of those of the B-OLED (3.1 V, 11260 cd m$^{-2}$, 6.23%) and Y-QLED (2.2 V, 21716 cd m$^{-2}$, 11.86%), indicating that both devices are effectively connected by the IZO. As shown in Supplementary Fig. 5, the EQE of the white LED can be further improved to 26.02% (power efficiency increased from 12.16 to 20.31 lm W$^{-1}$) by replacing the FL B-OLED with mCPPO1:Firpic-based Ph B-OLED. The emission spectra of the white LED are shown in Fig. 4d, which are normalized to the peak of the R emission. Because the B and the G emission are turned-on at a relatively high voltage, their emission intensity are gradually enhanced when the driving voltage is increased. As a result, the color coordinates are shifted from (0.42, 0.33) to (0.31, 0.36) when the brightness is increased from 1000 to 107,000 cd m$^{-2}$, as shown in Supplementary Fig. 6. Conventionally, it is very challenging to realize a color-stable white LED due to the migration of the exciton recombination zone as well as the difference of the driving voltage of R, G, and B emission[41–44]. Fortunately, with our novel tandem structure, we are able to improve the color stability by using an AC driving. Figure 4h shows the driving

**Table 1 The performances of the multifunctional tandem LED.**

| Function | Connection | Color | Voltage (V) | Brightness (cd m$^{-2}$) | Maximum EQE (%) | Color coordinate |
|---|---|---|---|---|---|---|
| Color tunable | Parallel | Red | 0 to −2.6 | 0–348 | 8.2 | (0.67, 0.32)–(0.60, 0.39) |
| | | Green | −5 to −8 | 15,800–51,590 | 10.5 | (0.30, 0.65)–(0.26, 0.69) |
| | | Blue | 0–8 | 0–9359 | 5.6 | (0.15, 0.30) |
| White emission | Series | White with FL-OLED | 8–15 | 2696–107,389 | 16.9 | (0.42, 0.33)–(0.31,0.36) |
| | | White with Ph-OLED | 7–15 | 364–58,605 | 26.02 | (0.23, 0.29)–(0.31, 0.36) |
| | Parallel | White with stable color | $V_Y = -12$ $V_B = 5-13$ | 1000–50,000 | Y-QLED: 11.86 B-OLED: 6.24 | (0.34, 0.36) |

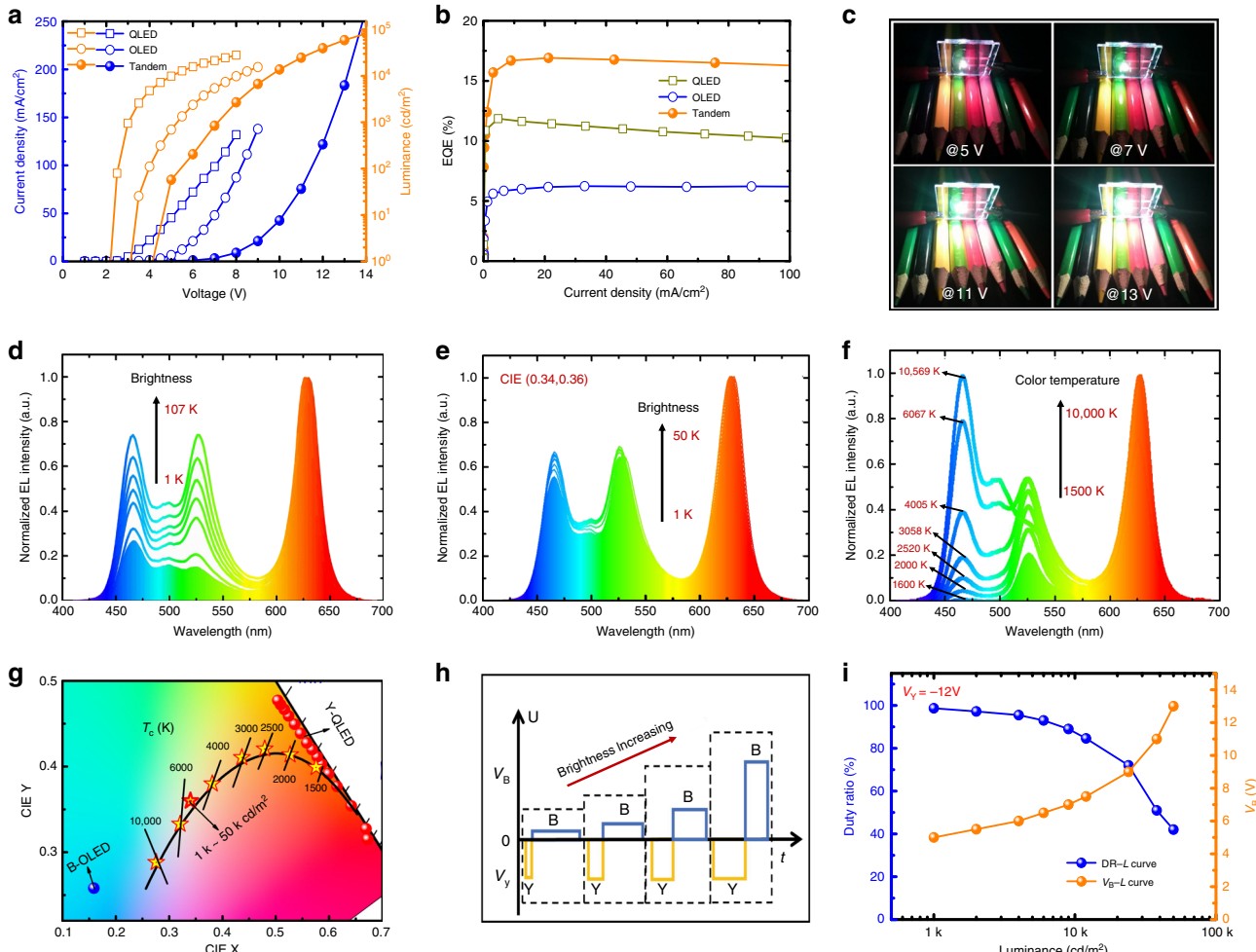

**Fig. 4 Performance of the tandem white LED. a** The *J–V–L*, **b** EQE–*J* characteristics of the B-OLED, Y-QLED, and tandem white LED. **c** The photos of an operating tandem white LED driven by different voltage levels. **d** Normalized emission spectra of the tandem white LED under **d** DC driving, and **e, f** AC driving. **g** CIE coordinates of the white LED (star), B-OLED (blue point), and Y-QLED (red point). Under AC driving, stable white emission with CIE coordinates fixed at (0.34, 0.36) can be obtained for a wide range of brightness (1000–50,000 cd m$^{-2}$). Also, the color temperature can be tuned from 1500 to 10,000 K, which well traces the blackbody locus. **h** AC signals and **i** DR–*L* and $V_B$–*L* curves for driving a color stable white LED.

signals for a color stable white LED. The $V_Y$ for controlling the Y-QLED is fixed at −12 V to ensure a stable Y emission. In this case, the brightness of the Y-QLED and B-OLED is controlled by the DR of the AC signals and the $V_B$, respectively. By gradually increasing $V_B$ and decreasing DR (Fig. 4i), a very stable white emission with a brightness increasing from 1000 to 50,000 cd m$^{-2}$ is obtained. As shown in Fig. 4g, the color coordinates are fixed at (0.34, 0.36) for a wide range of brightness. The corresponding emission spectra are shown in Fig. 4e. Because the QDs have relatively narrow emission spectra and the emission of G-QDs

peaked at 524 nm is optimized for display purpose, the resultant white emission exhibits a low color rendering index (CRI) of only 60, which could be further improved by using G-QDs with redder emission (e.g. 550 nm). Besides stable white emission for a wide range of brightness, the white LED can also be tuned to mimic the blackbody radiation. As shown in Fig. 4f, by varying the $V_Y$ and the $V_B$, the emission intensity of Y-QLED and B-OLED can be tuned in a wide range, thereby allowing us to obtain a series of white emission with different ratios of yellow and blue emission intensity. Specifically, the color coordinates of the white LED can

be tuned to trace the blackbody locus for a wide range of correlated color temperature (1500–10,000 K), as shown in Fig. 4g. The detailed performances of the white LED are shown in Table 1. The white LED, with high efficiency, high brightness, pure white color, high color stability, and tunable color temperature, could be the ideal lighting source for high-quality illumination, as demonstrated in Fig. 4c.

## Discussion

In summary, a multifunctional hybrid tandem LED is demonstrated, which is realized by: (1) stacking a Y-QLED and a B-OLED vertically, (2) using a conductive and transparent IZO as the ICE, and (3) configuring the EML of Y-QLED with mixing R-QDs and G-QDs. With the novel IZO ICE, the Y-QLED and B-OLED can be effectively connected either in parallel or in series, thus enabling the tandem LED to work multi-functionally. Under parallel connection, the tandem LED can be tuned to emit a wide range of colors, including R, G, and B primary colors, as well as arbitrary colors that are inside a color triangle defined by the primary colors. The achievable brightness of the R, G, and B primary colors could meet the requirement of the display application. With the color-tunable tandem LED, a full-color display with a color gamut up to 63% NTSC could be realized, and also, the absorptive CFs could be eliminated, the problematic EML patterning could be canceled and the pixel density could be improved by threefold. When under conventional series connection, the tandem LED exhibits an efficient white emission with a high brightness of 107,000 cd m$^{-2}$ and an EQE up to 26.02%. Under AC driving, stable white color (0.34, 0.36) over a wide range of brightness (1000–50,000 cd m$^{-2}$), as well as tunable color temperature (1500–10,000 K) can be achieved. We believe that our demonstrated two-terminal tandem LED, with multi-functionality of full-color-tunability and white light emission, could find potential applications in both full-color display and solid-state lighting.

## Methods

**Device structures**. Multifunctional tandem LEDs were fabricated by using the structure of glass/ITO/PEDOT:PSS (45 nm)/TFB (40 nm)/R-QDs:G-QDs (18 nm, 1:20 for color-tunable device or 1:9 for white device)/ZnMgO (40 nm)/Al (2 nm)/ IZO (60 nm or 80 nm)/MoO$_3$ (8 nm)/NPB (45 nm)/MADN:DSA-Ph (25 nm)/TPBi (40 nm)/LiF (1 nm)/Al (100 nm) (abbreviations: PEDOT:PSS = poly(3,4-ethylenedioxythiophene)/polystyrenesulfonate, TFB = poly[(9,9-dioctylfluorenyl-2,7-diyl)-co(4,4'-(N-(p-butylphenyl))diphenylamine)], NPB = N,N'-bis-(1-naphthyl)-N,N'-diphenyl-1,1'-biphenyl-4,4''-diamine, MADN = 2-methyl-9,10-di(2-naphthyl) anthracene, DSA-Ph = p-bis(p-N,N-diphenyl-aminostyryl)benzene, TPBi = 2,2',2''-(1,3,5-benzinetriyl)-tris(1-phenyl-1-H-benzimidazole).

To improve the efficiency of white device, the phosphorescent B-OLED-based tandem LED with structure of glass/ITO/PEDOT:PSS (45 nm)/TFB (40 nm)/R-QDs:G-QDs (18 nm, 1:9)/ZnMgO (40 nm)/Al (2 nm)/IZO (60 nm or 80 nm)/ HATCN (20 nm)/TAPC (40 nm)/mCP (5 nm)/Firpic:mCPPO1 (30 nm)/TmPyPB (30 nm)/LiF (1 nm)/Al (100 nm) was also fabricated. (abbreviations: HATCN = dipyrazino[2,3-f:2',3'-h]quinoxaline-2,3,6,7,10,11-hexacarbonitrile, TAPC = 1,1-bis [4-[N,N-di(p-tolyl) aminophenyl]cyclohexane, mCP = 1,3-bis(9-carbazolyl) benzene, Firpic = Ir(4,6-dFppy)2(pic), mCPPO1 = 9-(3-(9H-carbazole-9-yl) phenyl)-3-(dibromophenylphosphoryl)-9H-carbazole, TmPyPB = 1,3,5-Tri(m-pyridin-3-ylphenyl)benzene).

**Device fabrication**. The cleaned ITO glass substrates with a sheet resistance of 25 Ω □$^{-1}$ were treated with an O$_2$ plasma for 10 min. After O$_2$ plasma treatment, the hole injection layer was deposited by spin-casting a PEDOT:PSS (Clevios AI 4083) at 3000 r.p.m. for 45 s and baked at 130 °C for 15 min on a hot plate. Then, the samples were transferred to a nitrogen-filled glovebox to sequentially fabricate the following layers. The hole transport layer was deposited by spin-casting a TFB solution (8 mg mL$^{-1}$ in chlorobenzene) at 3000 r.p.m. for 40 s followed by annealing at 110 °C for 15 min. The QD EML was formed by spin-casting a mixed solution of R- and G-QDs (R-QDs: CdZnSe/ZnS/OT, core and shell ~12.2 nm; G-QDs: CdZnSeS/ZnS/oleic acid, core and shell ~11.6 nm. The QDs were purchased from Suzhou Mesolight Inc.) at 3000 r.p.m. for 40 s, followed by baking at 100 °C for 5 min. The mixed QD solution was obtained by mixing a R-QD solution (10 mg mL$^{-1}$ in octane) with a G-QD solution (10 mg mL$^{-1}$ in octane). The mixing ratio of R- and G-QDs can be tuned by varying the volume ratio of both

solutions. After EML fabrication, the electron transport layer was deposited by spin-casting the Zn$_{0.85}$Mg$_{0.15}$O nanoparticles solution (20 mg mL$^{-1}$ in ethanol) at 2500 r.p.m. for 40 s and backed at 100 °C for 10 min.

After that, the samples were transferred to a high-vacuum evaporation chamber to deposit a 2 nm Al protective layer at a base pressure of 5 × 10$^{-4}$ Pa. Then, the IZO with a thickness of 60 nm for color-tunable device or 80 nm for white tandem device was fabricated by a magnetron sputtering system at a working pressure of 0.45 Pa, a power of 50 W, an Ar flow of 20 sccm. The IZO target is composed of 90 wt% In$_2$O$_3$ and 10 wt% ZnO. The sputtered 80 and 60 nm IZO films show the sheet resistance of ~87 Ω □$^{-1}$ and ~116 Ω □$^{-1}$, respectively. The IZO was patterned by using a shadow mask. Even with a 2-nm-thick Al, the Al/IZO electrode still exhibits a high transparency of 85% and a low reflectance of 8%, as shown in Supplementary Fig. 7. After IZO fabrication, the samples were transferred to a high-vacuum evaporation chamber to deposit the OLED at a base pressure of 5 × 10$^{-4}$ Pa. For fluorescent B-OLED, the layers of MoO$_3$ (8 nm), NPB (45 nm), MADN:DSA-Ph (12%, 25 nm), TPBi (40 nm), LiF (1 nm), and Al (100 nm) were sequentially deposited, and for phosphorescent B-OLED, the layers of HATCN (20 nm), TAPC (40 nm), mCP (5 nm), Firpic:mCPPO1 (8%, 30 nm), TmPyPB (30 nm), LiF (1 nm), and Al (100 nm) were sequentially deposited. The layout of the tandem LED with bottom ITO, top Al, and intermediate IZO electrodes are shown in Supplementary Fig. 8.

**Device characterization**. The thicknesses of IZO films and all the solution-processed films were measured by using a Bruker DektakXT Stylus Profiler. The AC square-wave voltage signals were provided by a waveform generator (JunCe JDS-6600). The evaporation rates and the thicknesses of all the organic layers, MoO$_3$, LiF, and Al were in situ monitored by a quartz crystal microbalance. The emission spectra of all the devices were measured by a PR670 spectrometer. The J–V–L characteristics were characterized by a programmable source meter (Keithley 2614B) and a PR670 spectrometer. Because of the high transparency of the Al/IZO ICE (Supplementary Fig. 7), the microcavity effect is quite weak. Considering the weak microcavity effect, the EQE was calculated by assuming that the emission is Lambertian. The calculated EQE was further verified by a method recommended by SR Forrest[45]. A large-area (613 mm$^2$) PIN-25D silicon photodiode was placed in close contact with the devices (active area 2 mm × 2 mm). By measuring the photodiode current and the driving current, the EQE (as photons per electron) was calculated by converting the photodiode current to emitted photons and the driving current to electrons.

## Data availability

The data that support the findings of this study are available from the corresponding author upon reasonable request.

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

## Acknowledgements

This work was supported by the National Natural Science Foundation of China (61775090), the Guangdong Natural Science Funds for Distinguished Young Scholars (2016A030306017), and the Guangdong Special Funds for Science and Technology Development (2017A050506001).

## Author contributions

S.C. conceived the idea, supervised the work, and wrote the paper. H.Z. conducted the experiments, collected the data, and drew the figures. S.C., H.Z., and Q.S. discussed the results and commented on the manuscript.

## Competing interests

The authors declare no competing interests.

## Additional information

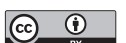

