## [Peer Review File · Nature Communications]

Reviewer #1 (Remarks to the Author):

The manuscript "Quantum-dot and organic hybrid tandem light-emitting diodes with multi-functionality of full-color-tunability and white-light-emission" is a very interesting piece of work on the preparation of a tunable LED by the use of a tandem structure combining a red-green LED using QDs as light emitters and an organic LED for the blue emission. The device can work in series and in parallel. When it is working in parallel a fine control of the color can be obtained. The system also offer a high EQE and white emission with fixed CIE coordinated for a broad range of brightness. I consider that the manuscript will be of the interest for Nature Communications readers. However, there are some points that authors should revise:

- There is no reference to the kind of QDs employed until the experimental section at the end of the manuscript. Table 1, introduces two kind of devices (FL-OLEDs and Ph-OLEDs) but the meaning and the differences between them are explained later. I suggest to introduce all the materials, systems and configurations in the first part of the manuscript to make it more easy reading.
- The preparation of QDs is not explained in the experimental section .
- Page 7: "To the best of our knowledge, this is the first demonstration of a single two-terminal device with full-color-tunability (also demonstrated in Supplementary Movie)." The full color tunability is obtained with the parallel configuration that is a three terminal configuration. Consequently, I suggest to remove this sentence .
- What is the meaning of the super index in the color coordinate of the white with FL-OLED in Table 1?
- For a more appropriated discussion of the white emission I suggest to include the color rendering index (CRI) obtained for the white emission.

Reviewer #2 (Remarks to the Author):

Authors reported the fabrication of hybrid LEDs consisting of green, red QD and blue organic molecules in the frame of tandem structure having IZO-based intermediate connecting electrode (ICE) between two units. Such hybrid tandem LEDs worked multifunctionally through selectively connecting in parallel and in series with AC and DC driving, respectively, enabling full-color tunability and white light emission for possible target applications to display and general lighting devices. In particular, this work offers a possibility that the generation of respective primary colors from white (or R/G/B mixing) pixel mode can be realized without use of color filters by manipulating AC driving parameters, though the brightness of R emission was very weak compared to other two colors. Given a novel device approach and significant device results, I recommend this paper for the publication to Nature Communications. Here are some minor points needed to be addressed.

- (1) Specific chemical composition and resistivity (or sheet resistance) of sputter-deposited IZO (ICE) are needed to be presented for general readers' information.
- (2) Compared to Figure 4(e), the description of Figure 4(f) was insufficient in the text. Authors are suggested to additionally state how (or which AC operating conditions) the results of Figure 4(f) were obtained.
- (3) White spectral evolution tendency with increasing brightness between hybrid devices integrated with fluorescent B-OLED (Figure 4(d)) versus phosphorescent B-OLED (Figure S5(c)) is different. What caused this?

Reviewer #3 (Remarks to the Author):

Zhang, Su , and Chen present in their manuscript "Quantum-dot organic hybrid tandem light-emitting diodes with multi-functionality of full-color-tunability and white light emission" a structure

of a tandem hybrid LED consisting of a blue organic LED and a red/green quantum dot LED. Depending on the driving scheme, they show that they can use their device either as white LED (for DC driving) or as color tunable device (using ac driving). To tune the color, they employ the voltage dependency of the emission color in the red/green quantum dot unit.

The manuscript is very well written and the results are interesting and impactful. Although neither the concept of color tuning in tandem OLEDs (see e.g. ref 34 of the manuscript) nor the shift of emission color with applied voltage is entirely new, the combinations of these effects and the fact that the authors can tune the color in a wide color space is impressive.

Overall, I suggest to publish the manuscript, once the following comments are taken into consideration:

- One shortcoming of this device is the problem to independently tune the brightness of emission. The authors state that they can tune the brightness by increasing or decreasing the length of the voltage pulse applied to the QD LED unit. Although this is certainly correct, this approach is limited. Can the authors estimate the maximum brightness of green emission and compare it to the maximum brightness of red emission?
- The tandem structure has a complicated micro cavity, in particular due to the thin Al in the middle of the device. Can the authors discuss their optimization of the cavity, i.e. how they chose the position of the emitter layers? If not optimized, this information will give the reader more information to judge the potential maximum efficiency of the device.
- please state the lifetime of the devices
- please add more details how the efficiency (EQE) of the devices is calculated. Do the authors assume a lambertian emission characteristic? Or is the angular dependency of emission taken into account? Given the structure of the device, I would assume a deviation from a purely lambertian emission.
- To compare the white light emission to literature values, please provide the luminous efficacy, i.e. lm/W and the CRI value.
- Did the authors measure the efficiency of their devices during AC driving as well and do they expect a difference to DC efficiencies?

Responses to reviewer #1

Q0: The manuscript “Quantum-dot and organic hybrid tandem light-emitting diodes with multi-functionality of full-color-tunability and white-light-emission” is a very interesting piece of work on the preparation of a tunable LED by the use of a tandem structure combining a red-green LED using QDs as light emitters and an organic LED for the blue emission. The device can work in series and in parallel. When it is working in parallel a fine control of the color can be obtained. The system also offer a high EQE and white emission with fixed CIE coordinated for a broad range of brightness. I consider that the manuscript will be of the interest for Nature Communications readers. However, there are some points that authors should revise:

A0: Thanks for your efforts on reviewing this manuscript. We sincerely appreciate your positive comments and valuable suggestions.

Q1: There is no reference to the kind of QDs employed until the experimental section at the end of the manuscript. Table 1, introduces two kind of devices (FL-OLEDs and Ph-OLEDs) but the meaning and the differences between them are explained later. I suggest to introduce all the materials, systems and configurations in the first part of the manuscript to make it more easy reading.

A1: Thanks for your valuable suggestions. Following your suggestions, we have modified the first sentence (highlighted) as “CdSe-based quantum-dot light-emitting diodes (QLEDs)”. Also we have defined the meaning of FL-OLEDs as “fluorescent organic light-emitting diodes (FL-OLEDs)” in the second paragraph on page 2. To clearly introduce all the materials, systems and the configuration, we have modified the first sentence in the results section on page 4 as “As shown in Figure 1, the multifunctional tandem LED consists of a bottom CdSe-based Y-QLED and a top MADN:DSA-Ph-based fluorescent (FL) B-OLED, which are vertically stacked and connected through an ICE. To obtain higher efficiency, the FL B-OLED can be replaced by a mCPPOI:FirPic-based phosphorescent (Ph) B-OLED.”

Q2: The preparation of QDs is not explained in the experimental section.

A2: Thanks for your kind reminder. Our group does not make the materials. All materials are commercially available. As we stated in the experimental section on page 14 “The QDs were purchased from Suzhou Mesolight Inc.”

Q3: Page 7: “To the best of our knowledge, this is the first demonstration of a single two-terminal device with full-color-tunability (also demonstrated in Supplementary Movie).” The full color tunability is obtained with the parallel configuration that is a three terminal configuration. Consequently, I suggest to remove this sentence.

A3: Thanks for your valuable suggestions. Following your suggestions, we have modified the sentence as “To the best of our knowledge, this is the first demonstration of a full-color-tunable device that is simply driven by a single two-terminal source (also demonstrated in Supplementary Movie).”

Q4: What is the meaning of the super index in the color coordinate of the white with FL-OLED in Table 1?

A4: Thanks for your kind reminder. The redundant super index has been removed.

Q5: For a more appropriated discussion of the white emission I suggest to include the color rendering index (CRI) obtained for the white emission.

A5: Thanks for your valuable suggestion. Because the spectra of the QDs are too narrow, the CRI of the white emission is only ~60. We have added a few sentences on page 11 “Because the QDs have relatively narrow emission spectra and the emission of G-QDs peaked at 525 nm is optimized for display purpose, the resultant white emission exhibit a low color rendering index (CRI) of only 60, which could be further improved by using G-QDs with redder emission (for example 550 nm).”

Responses to reviewer #2

Q0: Authors reported the fabrication of hybrid LEDs consisting of green, red QD and blue organic molecules in the frame of tandem structure having IZO-based intermediate connecting electrode (ICE) between two units. Such hybrid tandem LEDs worked multifunctionally through selectively connecting in parallel and in series with AC and DC driving, respectively, enabling full-color tunability and white light emission for possible target applications to display and general lighting devices. In particular, this work offers a possibility that the generation of respective primary colors from white (or R/G/B mixing) pixel mode can be realized without use of color filters by manipulating AC driving parameters, though the brightness of R emission was very weak compared to other two colors. Given a novel device approach and significant device results, I recommend this paper for the publication to Nature Communications. Here are some minor points needed to be addressed.

A0: Thanks for your efforts on reviewing this manuscript. We do thanks for your positive comments and helpful suggestions.

Q1: Specific chemical composition and resistivity (or sheet resistance) of sputter-deposited IZO (ICE) are needed to be presented for general readers' information.

A1: Thanks for your kind reminder. Following your suggestions, we have added a sentence in the experimental section on page 15, the sentence "The IZO target is composed of 90 wt% In₂O₃ and 10 wt% ZnO. The sputtered 80 nm and 60 nm IZO films show the sheet resistance of ~87 Ω/□ and ~116 Ω/□, respectively."

Q2: Compared to Figure 4(e), the description of Figure 4(f) was insufficient in the text. Authors are suggested to additionally state how (or which AC operating conditions) the results of Figure 4(f) were obtained.

A2: Thanks for your good suggestions. Following your suggestions, we have added a

few sentences on page 12 “Besides stable white emission for a wide range of brightness, the white LED can also be tuned to mimic the black-body radiation. As shown in Figure 4 (f), by varying the V_Y and the V_B , the emission intensity of Y-QLED and B-OLED can be tuned in a wide range, thereby allowing us to obtain a series of white emission with different ratios of yellow and blue emission intensity. Specifically, the color coordinates of the white LED can be tuned to trace the blackbody locus for a wide range of correlated color temperature (1500 ~10000 K), as shown in Figure 4 (g). ”

Q3: White spectral evolution tendency with increasing brightness between hybrid devices integrated with fluorescent B-OLED (Figure 4(d)) versus phosphorescent B-OLED (Figure S5(c)) is different. What caused this?

A3: As shown in the below figure, which is exactly the same as Figure S5 (c) except that it is normalized to the red emission for clear comparison, the blue emission is gradually decreased when the brightness is increased, which is opposite to the tendency shown in Figure 4 (d). This difference is caused by the roll-off behaviors of fluorescent B-OLED and phosphorescent B-OLED. The efficiency of phosphorescent B-OLED decreases more quickly, thereby leading to the reduction of blue emission at high brightness.

Responses to reviewer #3

Q0: Zhang, Su , and Chen present in their manuscript "Quantum-dot organic hybrid tandem light-emitting diodes with multi-functionality of full-color-tunability and white light emission" a structure of a tandem hybrid LED consisting of a blue organic LED and a red/green quantum dot LED. Depending on the driving scheme, they show that they can use their device either as white LED (for DC driving) or as color tunable device (using ac driving). To tune the color, they employ the voltage dependency of the emission color in the red/green quantum dot unit.

The manuscript is very well written and the results are interesting and impactful. Although neither the concept of color tuning in tandem OLEDs (see e.g. ref 34 of the manuscript) nor the shift of emission color with applied voltage is entirely new, the combinations of these effects and the fact that the authors can tune the color in a wide color space is impressive.

Overall, I suggest to publish the manuscript, once the following comments are taken into consideration:

A0: Thanks for your efforts on reviewing this manuscript. We do thanks for your valuable suggestions and positive comments.

Q1: One shortcoming of this device is the problem to independently tune the brightness of emission. The authors state that they can tune the brightness by increasing or decreasing the length of the voltage pulse applied to the QD LED unit. Although this is certainly correct, this approach is limited. Can the authors estimate the maximum brightness of green emission and compare it to the maximum brightness of red emission?

A1: We absolutely agree with your opinion that the brightness of the red and the green emission cannot be independently tuned. As shown in Figure 3 (c) and Table 1, the maximum brightnesses of the red and the green emission are 0~347 and 12440~45450 cd/m², respectively. For display application, the brightness of the green emission is too high; however, as stated in the manuscript, the brightness

of the green could be further reduced to an application level by decreasing the length of the voltage pulse.

Q2: The tandem structure has a complicated micro cavity, in particular due to the thin Al in the middle of the device. Can the authors discuss their optimization of the cavity, i.e. how they chose the position of the emitter layers? If not optimized, this information will give the reader more information to judge the potential maximum efficiency of the device.

A2: We thank the reviewer for this good question. There indeed exists the microcavity effect in our tandem device; however, the microcavity effect is quite weak due to the high transparency of the Al/IZO electrode. As shown in below Figure (also included as Figure S7 in the revised manuscript), the Al (2 nm)/IZO (60 nm) exhibits a high transparency of 85% and a low reflectance of 8%.

Due to the weak microcavity effect, the emission is only slightly modified by the cavity. As shown in Figure S4, when the IZO thickness is increased from 40 to 80 nm, the emission of the Y-QLED are slightly red-shifted due to the longer cavity. Therefore, as we stated on page 7 “The purity of G is also affected by the thickness of IZO due to the microcavity effect, as shown in Figure S4. At an optimal R-QDs: G-QDs mixing ratio of 1:20 and an IZO thickness of 60 nm, the purest G emission with a color coordinate of (0.26, 0.29) is obtained.” Although the device with 40 nm IZO also shows the pure G emission, the higher sheet resistance of the thinner IZO is detrimental to the device performance.

Figure S7. The transmittance and reflectance spectra of the Al(2 nm)/IZO(60 nm) film.

Q3: please state the lifetime of the devices

A3: Thanks for your valuable suggestions. As we stated on page 2 “One reason is that the B-QLEDs are unstable, with a short T50 lifetime of ~200 h at an initial brightness of 1000 cd/m² [3,21], which lags far behind those of R- and G-QLEDs. While the stability of B-QLEDs is being questioned, its organic cousins – the fluorescent (FL) organic (O) LEDs, are relatively stable and have been applied in displays for years. By substituting the B-QLEDs with B-OLEDs, a hybrid display could be realized, which can enjoy both the high saturation of QLEDs as well as the high stability of OLEDs.” Therefore, as the unstable B-QLED is replaced by highly stable B-OLED in our tandem device, we believe that the lifetime of our devices can be improved.

In addition, as pointed out in literature (Appl. Phys. Lett. 1996, 69, 2160 – 2162. Jpn. J. Appl. Phys. 2000, 39, 3463 – 3465. Organic Electronics 2013, 14, 2001 – 2006.), the AC driving is good for improving the stability of the devices. Because the charges accumulated during forward driving could be driven out by the negative driving.

Q4: please add more details how the efficiency (EQE) of the devices is calculated. Do the authors assume a lambertian emission characteristic? Or is the angular dependency of emission taken into account? Given the structure of the device, I

would assume a deviation from a purely lambertian emission.

A4: Thanks for your good suggestions and questions. As we answered in A2, although there exists a microcavity effect, it is quite weak due to the high transparency of the Al/IZO electrode (Figure S7). The balanced white emission shown in Figure 4 (e) also implies the microcavity effect is weak, otherwise it is difficult to obtain balanced white emission due to the mode selection of the cavity.

As we stated in the experimental section on page 16 “The current density–luminance–voltage (J–V–L) characteristics were characterized by a programmable source meter (Keithley 2614B) and a PR670 spectrometer. Because of the high transparency of the Al/IZO ICE (Figure S7), the microcavity effect is quite weak. Considering the weak microcavity effect, the EQE was calculated by assuming that the emission is lambertian. The calculated EQE was further verified by a method recommended by SR Forrest⁴⁵. A large area (613 mm²) PIN-25D silicon photodiode was placed in close contact with the devices (active area 2 mm × 2 mm). By measuring the photodiode current and the driving current, the EQE (as photons per electron) was calculated by converting the photodiode current to emitted photons and the driving current to electrons.”

Q5: To compare the white light emission to literature values, please provide the luminous efficacy, i.e. lm/W and the CRI value.

A5: Thanks for your good suggestions. Following your suggestions, we have added a few sentences on page 11 “ (power efficiency increased from 12.16 to 20.31 lm/W) ” and “Because the QDs have relatively narrow emission spectra and the emission of G-QDs peaked at 524 nm is optimized for display purpose, the resultant white emission exhibits a low color rendering index (CRI) of only 60, which could be further improved by using G-QDs with redder emission (for example 550 nm).”

Q6: Did the authors measure the efficiency of their devices during AC driving as well and do they expect a difference to DC efficiencies?

A6: Under AC driving, the B-OLED and Y-QLED are alternately turned on; we are unable to measure the instant efficiency of both devices. However, we did measure the static efficiency of both devices. As shown in Figure 3 (a), the B-OLED and Y-QLED exhibit a high EQE of 5.6% and 11.2%, respectively. We believe that the instant efficiency (AC driving)of both devices could be higher than their static efficiency (DC driving) due to better heat dissipation and removal of accumulated charges.

REVIEWERS' COMMENTS:

Reviewer #1 (Remarks to the Author):

Authors have revised properly the manuscript taking into account the suggestions of the reviewers. I suggest the publication on the revised version .

Reviewer #2 (Remarks to the Author):

The authors satisfactorily addressed all the points raised and the manuscript can be accepted .

Reviewer #3 (Remarks to the Author):

The authors have addressed all my comments and I suggest publishing the manuscript.